# Are Humanized Mouse Models Useful for Basic Research of Hepatocarcinogenesis through Chronic Hepatitis B Virus Infection?

**DOI:** 10.3390/v13101920

**Published:** 2021-09-24

**Authors:** Masataka Tsuge

**Affiliations:** 1Natural Science Center for Basic Research and Development, Department of Biomedical Science, Research and Development Division, Hiroshima University, 1-2-3 Kasumi, Minami-ku, Hiroshima 734-8551, Japan; tsuge@hiroshima-u.ac.jp; Tel.: +81-82-257-1510; 2Department of Gastroenterology and Metabolism, Graduate School of Biomedical and Health Sciences, Hiroshima University, 1-2-3 Kasumi, Minami-ku, Hiroshima 734-8551, Japan; 3Research Center for Hepatology and Gastroenterology, Hiroshima University, 1-2-3 Kasumi, Minami-ku, Hiroshima 734-8551, Japan

**Keywords:** HBV, HCC, HBs, HBx, humanized mouse

## Abstract

Chronic hepatitis B virus (HBV) infection is a global health problem that can lead to liver dysfunction, including liver cirrhosis and hepatocellular carcinoma (HCC). Current antiviral therapies can control viral replication in patients with chronic HBV infection; however, there is a risk of HCC development. HBV-related proteins may be produced in hepatocytes regardless of antiviral therapies and influence intracellular metabolism and signaling pathways, resulting in liver carcinogenesis. To understand the mechanisms of liver carcinogenesis, the effect of HBV infection in human hepatocytes should be analyzed. HBV infects human hepatocytes through transfer to the sodium taurocholate co-transporting polypeptide (NTCP). Although the NTCP is expressed on the hepatocyte surface in several animals, including mice, HBV infection is limited to human primates. Due to this species-specific liver tropism, suitable animal models for analyzing HBV replication and developing antivirals have been lacking since the discovery of the virus. Recently, a humanized mouse model carrying human hepatocytes in the liver was developed based on several immunodeficient mice; this is useful for analyzing the HBV life cycle, antiviral effects of existing/novel antivirals, and intracellular signaling pathways under HBV infection. Herein, the usefulness of human hepatocyte chimeric mouse models in the analysis of HBV-associated hepatocarcinogenesis is discussed.

## 1. Introduction

Despite the global promotion of a universal vaccination program for hepatitis B virus (HBV) infection, an estimated 257 million people still suffer from chronic HBV infection and, each year, an estimated 887,000 individuals die from HBV-related liver diseases, including liver cirrhosis and hepatocellular carcinoma (HCC) [1]. Acute HBV infection via exposure to blood or other body fluids through sexual intercourse, unsafe injections, or injury with sharp instruments including medical devices is also recognized as a global problem. Therefore, vaccines are administered to adolescents besides their administration to newborns, expecting a long-term effect in preventing HBV infection in adults [2]. As pegylated interferons and nucleotide/nucleoside analogs have been approved for chronic hepatitis B treatment, it is now possible to strongly suppress HBV replication, resulting in the reduction in the severity of liver inflammation and fibrosis [3,4,5,6,7,8,9]. The current guidelines for managing chronic hepatitis B recommend long-term treatments using these antivirals, with the aim to prevent disease progression and improve patients’ quality of life [10,11,12]. However, as it is still difficult to eradicate HBV from hepatocytes with the current therapies, there is still a risk of HCC development, even when HBV replication is continuously suppressed with antiviral therapy. Therefore, to further reduce the incidence of HCC, it is necessary to develop novel antiviral drugs that can lead to viral eradication; however, most novel drugs are still in preclinical or phase 1 or 2 clinical trials [13].

Considering that HCC occasionally develops even when HBV replication is suppressed substantially by antiviral therapy, HBV-related proteins, such as hepatitis B surface (HBs), hepatitis B core (HBc), polymerase (pol), and hepatitis B x (HBx) protein, might be associated with carcinogenesis as their production is maintained during antiviral therapy. Therefore, to prevent hepatocarcinogenesis more effectively, it is important to understand the mechanisms of HBV-related hepatocarcinogenesis in detail and identify molecules that accelerate carcinogenesis by cooperating with viral proteins.

Recently, humanized mice with livers carrying transplanted human hepatocytes have been used as an experimental model for HBV infection and replication. As humanized mice are generated from severe immunodeficient mouse lines, hepatitis does not occur in HBV-infected mice. Therefore, using this mouse model, the direct effect of HBV infection in human hepatocytes can be analyzed without host immune responses. However, the association between HBV infection and hepatocarcinogenesis has not been analyzed using this mouse model as its lifespan is comparatively shorter than that of normal mice, and HCC does not occur in chimeric mouse livers. In this review, I discuss the usefulness of a humanized mouse model for analyzing hepatocarcinogenesis via HBV infection.

## 2. HBV Infection Is Animal Specific

HBV is a member of the Hepadnaviridae family and contains a 3.2-kb partially double-stranded circular DNA genome in the viral particle. HBV attaches to heparan sulfate proteoglycan (HSPG) on the surface of hepatocytes [14,15,16,17], and then virions enter the hepatocytes through transfer to sodium taurocholate cotransporting polypeptide (NTCP) [18]. Although NTCP expression in the liver can be observed in various animals, including mice, HBV infection is limited to human primates [19]. As the myristoylated pre-S1 subdomain of the large HB protein, which is considered an important region for the attachment of the virus to hepatocytes, can bind to hepatocytes of various animals, including mice, regardless of their susceptibility to HBV [20,21], the restricted infectivity of HBV to primates and scandentia (treeshrews) is considered to be due to the post-binding steps, such as membrane fusion, and not the presence or absence of the binding receptor [20]. Owing to this species-specific liver tropism in the early stages of HBV infection, suitable animal models for analyzing HBV replication and developing antivirals are lacking.

## 3. Construction of a Human Hepatocyte Chimeric Mouse Model for Hepatitis Virus Infection

To construct suitable animal models for gene therapy applications and for the analyses of biological mechanisms in metabolic diseases, autologous hepatocellular transplantation has been performed in mice, rabbits, and dogs since 1990 [22,23,24]. However, a serious problem in these animal experiments was that the replacement rates of the transplanted hepatocytes in the host livers were very low (less than 1%). In 1994, Rhim et al. succeeded in improving the replacement rate of autologously transplanted liver cells by up to 80% using albumin-urokinase (Alb-uPA) transgenic mice in which the urokinase gene is driven by the murine albumin promoter/enhancer and accelerates hepatocyte death [25,26]. Furthermore, they succeeded in constructing a rat hepatocyte chimeric mouse model, in which mouse hepatocytes were replaced with transplanted rat hepatocytes, using Alb-uPA transgenic mice backcrossed with a nude mouse strain [27]. Thus, repopulation with xenogeneic hepatocytes in Alb-uPA transgenic mouse livers under immune-deficient conditions has been indicated to be feasible, especially the construction of human hepatocyte chimeric mice with livers carrying human hepatocytes.

In 2001, a humanized mouse model was developed using Alb-uPA/SCID mice, which was generated by backcrossing Alb-uPA transgenic mice with the severe combined immunodeficiency (SCID) mouse strain [28,29]. As the Alb-uPA/SCID mice are severely immunodeficient, most of their hepatocytes can be replaced with transplanted human primary hepatocytes without immunological elimination [29,30] and their liver tissues are susceptible to HBV [28,31] and hepatitis C virus (HCV) [29,32]. Currently, several humanized mouse models, using Alb-uPA Tg/*Rag2* KO mice, cDNA-uPA Tg/SCID mice, *Fah*^−/−^/*Rag2*^−/−^/*IL2rγ*^−/−^ (FRG) mice, *Fah*^−/−^/*NOD*/*Rag1*^−/−^/*IL2rγ*c^null^ (FNRG) mice, and herpes simplex virus type 1 thymidine kinase *NOD*/*SCID*/*IL2rγ*c^null^ (HSV-TK-NOG) mice, have been developed, and these models have demonstrated susceptibility to HBV [28,33,34,35,36,37,38,39,40]. Although these humanized mouse models are useful for analyzing the HBV life cycle [40,41,42,43,44,45,46,47] and antiviral effects of existing and novel antivirals [48,49,50,51,52,53,54,55,56,57,58,59,60], it has not yet been clarified whether hepatocarcinogenesis mechanisms can be analyzed using these mouse models.

Recently, dual chimeric mouse models that carry not only human hepatocytes but also human immune cells have been developed by transplanting both human hematopoietic stem cells (HSCs) and either adult or fetal hepatocytes into some immune-deficient mouse models, such as FRG mouse [42,61,62,63,64,65]. HBV infects these dual chimeric mice, and hepatitis and liver fibrosis can be observed after HBV inoculation. However, the replacement rates to human hepatocytes in the liver are still lower than that in human hepatocyte chimeric mice, and HCC development has not been reported in these mouse models. Therefore, the replacement rate should be improved in these models to analyze the mechanisms of HCC development. If the replacement rate in dual chimeric mouse models can be improved as much as that in human hepatocyte chimeric mouse model, comprehensive gene expression analyses can be performed using their livers. Furthermore, it might help clarify the differences in the effects of HBV infection and human immunity on HBV-related hepatocarcinogenesis.

## 4. Analyzing the Association between HBV Genotype and Hepatocarcinogenesis Using HBV-Infected Humanized Mouse Models

HBV is categorized into nine genotypes based on nucleotide differences, and the clinical features of chronic hepatitis B, such as the incidence of HCC, are partially different among HBV genotypes. A Taiwanese cohort study revealed that the incidence of HCC among HBV genotype C carriers was 2.35-fold higher than that among HBV genotype B carriers [66]. In contrast, early onset non-cirrhotic HCC is more common in patients with HBV genotype B infection than in patients with HBV genotype C infection [67,68,69]. Another study indicated that patients with HBV genotype C infection have a 4-fold higher risk of developing cirrhosis and HCC than those with HBV genotype A, B, or D infection [70]. Based on these clinical studies, HBV genotype C infection might induce hepatocarcinogenesis more commonly than the other HBV genotypes. Therefore, it should be possible to indicate the differences in HCC development among HBV genotypes using a humanized mouse model. Previously, we generated HBV genotype A- and genotype C-infected human hepatocyte chimeric mice and compared the mRNA expression profiles of human hepatocytes obtained from mouse livers by next-generation sequencing [71]. Although the regulated pathways were similar between HBV genotype A and C infections, the induction levels of HBV infection were different (Table 1). Notably, genes associated with oxidative stress and the Wnt signaling pathway, which are well known to induce carcinogenesis [72,73,74,75], were more highly induced in human hepatocytes with HBV genotype C infection than in those with HBV genotype A infection. As other studies have also indicated that cellular stresses, such as oxidative damage, in mouse livers with HBV genotype C2 and Ba infections were significantly higher than those in mouse livers with infection by other HBV genotypes [76,77], these results might reflect the differences in the incidence of HCC among HBV genotypes.

In addition, thymosin-β4 (TMSB4X) and glutamate-ammonia ligase (GLUL), which were significantly upregulated by HBV genotype C and A infections, respectively, have been reported to be associated with other cancers. Although it has been reported that intracellular β-thymosins regulate monomeric actin to control actin polymerization in cells and extracellular TMSB4X promotes corneal and dermal wound healing and cardiac repair after ischemic injury [78], TMSB4X upregulation is frequently observed during tumor progression and is associated with carcinogenesis and metastasis in various cancers [79,80,81,82]. However, TMSB4X upregulation in HCC tissues is not frequent [83] and the association between TMSB4X and HCC has not been fully analyzed. GLUL is an enzyme involved in the synthesis of glutamine that catalyzes the condensation of glutamate and ammonia in an ATP-dependent manner [84]. Glutamine dependency is considered to be enhanced in cancer cells, and increased glutamine catabolism in MYC-induced liver tumors is associated with GLUL downregulation [85]. GLUL expression in the livers with HBV genotype C infection is not upregulated compared with that in the livers with HBV genotype A infection (Table 1), suggesting an association between GLUL expression with the difference in HCC incidence among HBV genotypes. Regardless, a genome-wide association study has reported that *GLUL* haplotype might be associated with familial HBV-related HCC [86], and further analyses may clarify the contribution of GLUL toward hepatocarcinogenesis.

Hayashi et al. performed a cDNA microarray using humanized mouse livers infected with HBV genotype F1b obtained from young Alaskan native patients with HBV-related HCC. They demonstrated that five genes associated with cell proliferation or carcinogenesis, v-myc avian myelocytomatosis viral oncogene homolog (*MYC*), Grb2-associated binding protein 2 (*GAB2*), bradykinin receptor B2 (*BDKRB2*), follistatin (*FST*), and mitogen-activated protein kinase kinase kinase 8 (*MAP3K8*), were significantly upregulated in human hepatocytes infected with HBV genotype F1b compared with their expression in hepatocytes infected with other genotypes [87]. Furthermore, they identified that the incidence of HCC in Alaskan native patients with HBV genotype F1b infection was associated with core mutations, and they showed enhanced upregulation of these five genes by mutations in the basal core promoter and pre-core lesion of the HBV genome using a humanized mouse model.

Considering that the clinical features of chronic hepatitis B are partially different among HBV genotypes, molecular mechanisms driving genotypic characteristics, including the incidence of HCC, might be revealed using a humanized mouse model. Furthermore, the association between the identified molecular targets and host immune responses might be analyzed using other in vitro and in vivo models to clarify HBV-related hepatocarcinogenesis, such as dual chimeric mouse models carrying both human hepatocytes and human immune cells [42,61,62], as the severity of liver inflammation is different among HBV genotypes.

## 5. Analyzing the Association between Intracellular Signaling Pathways and Hepatocarcinogenesis Using HBV-Infected Humanized Mouse Models

It is well known that endoplasmic reticulum (ER) stress is associated with hepatocarcinogenesis. As ER stress leads to oxidative stress and DNA damage in hepatocytes, it can regulate intracellular signaling pathways related to cell proliferation, apoptosis, and inflammatory cytokine and chemokine production [71,88,89], resulting in carcinogenesis. In HBV infection, ER stress has been reported to be induced by calcium depletion in the ER via the accumulation of the HBx and HB proteins (Figure 1) [88,90,91,92]. HBx induces the unfolded protein response (UPR), leading to the activation of the activating transcription factor 6 and inositol-requiring enzyme 1/X-box binding protein 1 pathways in the UPR [93]. Cho et al. reported that HBx induces the proliferation of hepatocellular carcinoma cells via activator protein 1 (AP1) overexpression as a result of ER stress [94]. In contrast, Li et al. reported that HBx localizes in the ER lumen and relieves ER stress by directly binding to glucose-related protein 78, resulting in the prevention of HCC cell death and negative regulation of DNA repair [95]. Although these findings seem to be contradictory, the former phenomenon might be induced in normal hepatocytes to repair intracellular disorders, and the latter might occur in cancer cells for progressing or promoting carcinogenesis. HBx perturbs intracellular Ca^2+^ homeostasis and reduces the uptake of Ca^2+^ in the mitochondria [96]. In contrast, ER stress in hepatocytes is also induced by the accumulation of large HB proteins [90,91]. When HB proteins accumulate in the ER, the ER expands and releases Ca^2+^ into the cytoplasm, activating ER stress signaling [88,90,91,97].

Based on a gene expression analysis using HBV-infected humanized mouse livers, the production of interleukin (IL)-8 (*CXCL8*) mRNA in the liver tissues was significantly upregulated by HBV infection, and *CXCL8* transcriptional activation might be induced by large HB proteins [88]. Although it has been reported that IL-8 could be induced by HBV infection [98,99,100], the mechanism of IL-8 induction has not been clarified, and research using HBV-infected humanized mice may help clarify the mechanism of IL-8 induction. IL-8 suppresses intracellular immune responses induced by HBV infection; studies have also demonstrated an association between IL-8 and HCC development. Serum IL-8 is also associated with the clinical features of HCC, such as tumor grade, extrahepatic metastasis, and poor prognosis in patients with HCC [101,102], and IL-8 regulates tumor cell growth, angiogenesis, and metastasis in the liver [103,104]. However, it is unclear whether serum IL-8 concentration correlates with the production of IL-8 from the liver tissues. This point needs to be verified using HBV-infected humanized mice.

When ER stress is induced by Ca^2+^ depletion, stromal interaction molecule 1 (Stim1) is introduced into the ER lumen, and store-operated Ca^2+^ entry (SOCE), a cell membrane calcium transporter, is activated to maintain intracellular Ca^2+^ homeostasis. However, the effect of HBV infection on the expressions of Stim1 and SOCE components, such as the calcium release-activated calcium modulator (Orai) family, has not been clarified. According to gene expression analyses using HBV-infected humanized mouse livers, Orai1 and Orai2 expressions were not significantly altered at 56-day post-HBV infection, but increased at 238-day post-infection (Figure 2) [88,89]. Thus, HBx and HB might accumulate in the ER gradually following infection, and the ER stress signal might be activated to maintain intracellular homeostasis once viral proteins accumulate beyond a certain level.

The ataxia-telangiectasia mutated (ATM)/checkpoint kinase 2 (Chk2)/p53 signaling pathway is one of the checkpoint systems for DNA damage. When double-strand DNA breaks occur in normal cells, ATM and Chk2 stabilize p53 by phosphorylation, leading to cell cycle arrest [105,106]. However, the ATM/Chk2/p53 pathway is known to be impaired in several cancers [107]. Analyses of clinical HCC tissues [108] and hepatocytes obtained from 15-month-old HBV-transgenic mice [109] have shown that Chk2 expression is increased and that Chk2 mislocalizes within mitotic structures. Moreover, another study indicated that the HB protein inactivates the ATM/Chk2/p53 pathway by downregulating reticulon 3 (RTN3), which promotes tumor growth [110]. Although RTN3 interacts with Chk2 and the complex is recruited to the ER, its activation is dependent on the calcium concentration in the ER [110]. As mentioned above, as ER stress by Ca^2+^ depletion can be observed in humanized mouse livers, the association between the ATM/Chk2/p53 pathway and hepatocarcinogenesis might be analyzed in detail using HBV-infected humanized mice.

It has been demonstrated that long non-coding RNAs (lncRNAs) participate in many cellular processes [111]. One of the proliferating cell nuclear antigen (PCNA) pseudogenes, *PCNAP1*, is a lncRNA with 78% homology to the 3′-UTR of the PCNA transcript. Feng et al. demonstrated that PCNAP1 expression was upregulated in both humanized mouse livers after HBV infection and in HCC tissues [112]. They also indicated that *PCNAP1* increased PCNA expression by competing with miR-154, resulting in the promotion of tumor cell growth. Although the functional analyses of PCNAP1 were performed using hepatoma cell lines and clinical HCC tissues, the upregulation of PCNAP1 might also be observed in HBV-infected humanized mice. The humanized mouse model might be useful for analyzing alterations in gene or pseudogene expression.

## 6. Analyzing HBV Integration and Hepatocarcinogenesis Using HBV-Infected Humanized Mouse Models

As HBV integration has been identified in several genes, such as *TP53* (p53), *CTNNB1,* and the promoter of *TERT* [113,114,115,116], HBV integration has been considered to be a potent driver of HCC development [117]. However, the contribution of HBV integration to HCC development is controversial as HBV integration occurs in random sites in the host genome and seems to not drive clonal expansion in cancer lesions in genomic studies [116,118,119,120,121]. Therefore, the analysis of HBV integration using HBV-infected humanized mice is desirable. Recently, ultra-deep sequencing analysis was performed using humanized mouse livers, and data from several time points after HBV infection were obtained [121]. HBV integration in mouse livers increased at 4–7 weeks after HBV infection, similar to the increase in intracellular HBV DNA, and approximately 70% of HBV integrations were observed in mitochondrial DNA. Considering that 0.1% of HBV integrations were observed in mitochondrial DNA in clinical tissues, and that most humanized mice have a short lifespan, it might be difficult to analyze the association between HBV integration and hepatocarcinogenesis using HBV-infected humanized mouse models.

## 7. Analyzing Epigenetic Modifications and Hepatocarcinogenesis Using HBV-Infected Humanized Mouse Models

Epigenetic modifications are significantly associated with the genesis and progression of liver fibrosis. According to a genome-wide methylation analysis using peripheral blood mononuclear cells (PBMCs) obtained from patients with HBV-related liver diseases, the CpG methylation profiles in PBMCs might be associated with peripheral immune responses and might be useful as a biomarker to predict liver cirrhosis [122]. In hepatocytes, viral pathogens engage in numerous cellular events, including epigenetic modifications that promote tumorigenesis [123]. In HBV infection, aberrant DNA methylation is considered to be induced by HBV-related proteins, especially HBx [124,125,126,127]. Although DNA methylation is commonly observed in normal aging livers, aberrant methylation of CpG islands of genes is induced during liver inflammation and hepatocarcinogenesis [124,125,126]. To analyze the effect of HBV infection on DNA methylation, a methylated CpG island amplification microarray analysis was performed using HBV-infected humanized mouse livers [128]. The number of methylated genes in HCV mice was similar to or slightly higher than that in HCV-infected humanized mouse livers, and it varied with the infection term. Furthermore, in this study, 70% of methylated genes were extracted from both HBV- and HCV-infected livers, and HBV-specific DNA methylation was not identified.

Other epigenetic modifications, such as histone modifications, are recognized as key intracellular alterations associated with carcinogenesis [129,130,131,132,133,134,135,136]. HBx has been demonstrated to promote HBV-induced HCC pathogenesis by inducing histone modifications [137,138,139,140]. The cyclic AMP response element-binding protein (CREB)-binding protein (CREBBP)/p300 transcriptional regulatory complex induces histone acetylation by interacting with HBx, leading to the upregulation of the expression of genes associated with tumorigenesis [140]. HBx also recruits the mSin3A/histone deacetylase 1 (HDAC1) or HDAC1/Sp1 complex and contributes to the activation of hepatocarcinogenesis by suppressing tumor suppressor genes [137,138]. Furthermore, histone methyltransferases, such as SET and MYND domain-containing 3 and suppressor of variegation 3–9 homolog 1 (SUV39h1), are also known to be upregulated in HCC tissues [129,130,134,141], and their expression is enhanced by HBx [134,136]. In addition, HBx is involved, directly or indirectly, in the impairment of efficient homologous recombination and is known to induce genomic instability by the inhibition of histone H2B monoubiquitylation and by structural maintenance of chromosome 5/6 (Smc5/6) ubiquitination using a hijacked CRL4 complex [142]. Although SUV39h1 upregulation is indicated in the HBV-infected humanized mouse model [134], exhaustive analyses, such as chromatin immunoprecipitation sequencing, have not yet been performed using HBV-infected humanized mice. As epigenetic modifications are considered to occur from the early phase of carcinogenesis and to change with disease progression, exhaustive analyses might contribute to the identification of novel therapeutic targets for HCC.

## 8. Conclusions and Future Prospects

In this review, I have discussed the usefulness of humanized mouse models carrying human hepatocytes in the liver for analyzing HCC development. As these mouse models are derived from immunodeficient mouse strains, the potential host immune response to HBV infection is effectively eliminated, allowing researchers to study the direct effect of HBV infection on human hepatocytes. Thus, these mouse models might be useful for identifying therapeutic molecular targets. However, these models cannot be used to analyze the effect of inflammation on the liver. Hepatocarcinogenesis is considered to be supported by not only by HBV proteins, but also continuous liver inflammation. The association between the identified therapeutic molecular targets and host immune responses should be analyzed using other in vitro and in vivo models for clarifying HBV-related hepatocarcinogenesis. Recently, dual chimeric mouse models which carry not only human hepatocytes, but also human immune cells have been developed [42,61,62]. These models allow researchers to analyze the effects of both HBV infection and human immunity on HBV-related hepatocarcinogenesis. However, the differences in clinical parameters, such as HBV protein production levels, the duration of HBV infection, and the strength of the immune responses, should also be considered while interpreting the results.

## Figures and Tables

**Figure 1 viruses-13-01920-f001:**
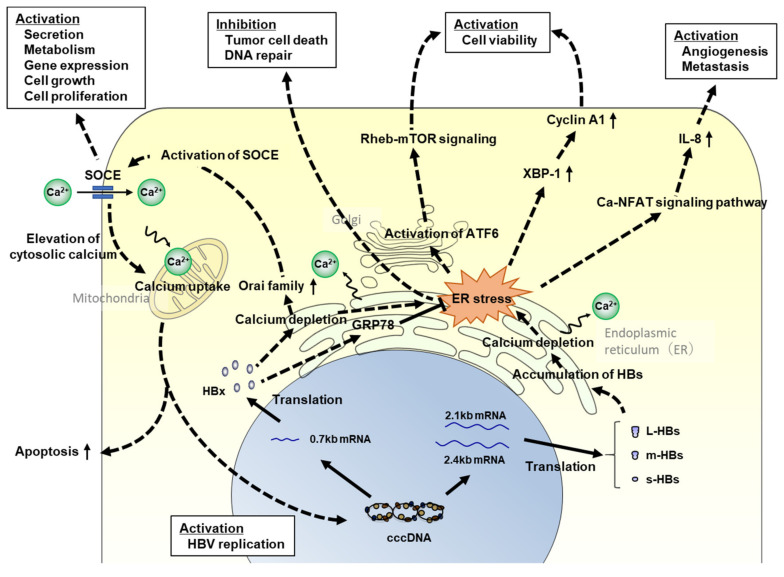
The association between HBV-related proteins and hepatocarcinogenesis via ER stress. Signaling pathways related to hepatocarcinogenesis via HBs or HBx protein-induced ER stress are shown. HBV, hepatitis B virus; cccDNA, covalently closed circular DNA; HBx, hepatitis B x protein; L-HBs, large hepatitis B surface protein; m-HBs, middle hepatitis B surface protein; s-HBs, small hepatitis B surface protein; ER, endoplasmic reticulum; SOCE, store-operated calcium entry; Orai, calcium release-activated calcium modulator; GRP78, glucose-related protein 78; Rheb, ras homolog enriched in brain; mTOR, mammalian target of rapamycin; XBP1, X-box binding protein; NFAT, nuclear factor of activated T cells; and IL-8, interleukin-8.

**Figure 2 viruses-13-01920-f002:**
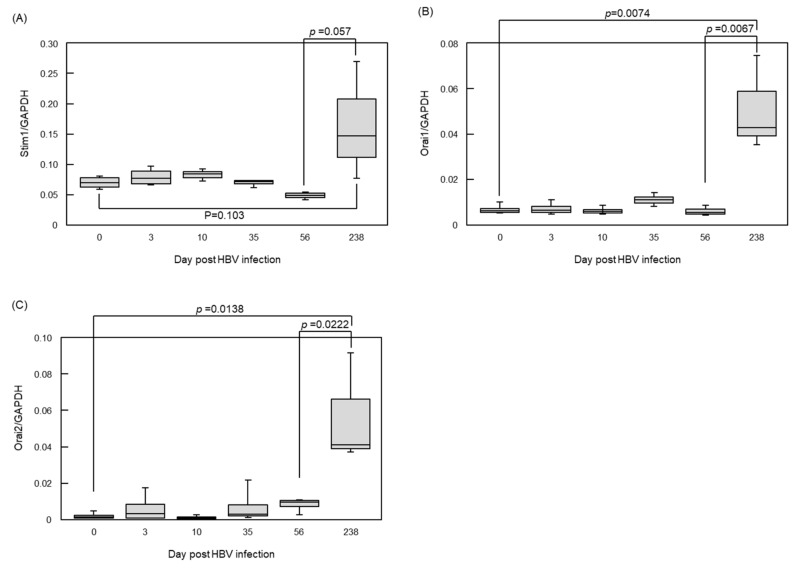
Upregulation of Orai family in human hepatocytes after long term HBV infection. Twenty-three human hepatocyte chimeric mice were prepared, and 19 were inoculated with 10^6^ copies of HBV via the mouse tail vein. Mice were sacrificed at 3, 10, 35, 56, and 238 days after inoculation, and human hepatocytes were collected from mouse livers obtained from HBV-infected or non-infected mice. Total RNA was extracted. Gene expression analysis was performed using next-generation sequencing, and Stim1 (**A**), Orai1 (**B**), and Orai2 (**C**) expression profiles were extracted. All statistical analyses were performed using the Student’s *t*-test. Stim1, stromal interaction molecule 1; Orai, calcium release-activated calcium modulator; *p*, *p* value.

**Table 1 viruses-13-01920-t001:** Comparison of induction rates of the top 10 genes which were upregulated in both HBV genotype A and C infection.

Gene	FC (Cont vs. GtA)	FC (Cont vs. GtC)	*p* Value (GtA vs. GtC)
*SAA1*	165.419	42.399	0.0418
*PRAP1*	4.7979	1.2349	0.0054
*LYZ*	283.224	44.5099	0.0144
*LCN2*	574.344	165.8939	0.0276
*SAA4*	6.877	2.583	0.0352
*RPL7A*	3.330	1.251	0.0014
*TMSB4X*	1.304	3.297	0.0005
*GLUL*	3.658	1.165	0.0004
*FGL1*	3.489	1.050	0.0010
*CD74*	14.546	3.923	0.0205
*CXCL10*	18.436	46.935	0.0157

Statistical analysis was performed by *t* test. FC, fold change; Cont, mice without HBV infection; GtA, mice with HBV genotype A infection; GtC, mice with HBV genotype C infection; SAA1, serum amyloid A1; PRAP1, proline-rich acidic protein 1; LYZ, lysozyme; LCN2, lipocalin 2; SAA4, serum amyloid A4; RPL7A, ribosomal protein L7a; TMSB4X, thymosin, beta-4; GLUL, glutamate ammonia ligase; FGL1, fibrinogen-like 1; and CXCL10, chemokine, CXC motif, ligand 10.

## Data Availability

Not applicable.

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
