# Peer review of "Are Humanized Mouse Models Useful for Basic Research of Hepatocarcinogenesis through Chronic Hepatitis B Virus Infection?"

_viruses, 2021, doi:10.3390/v13101920_

Round 1

Reviewer 1 Report

In this review article, Tsuge discussed the usefulness of humanized mouse model in basic research of HBV-related HCC. The author summarized the history of constructing the human hepatocyte chimeric mouse model and potential studies of HBV-related HCC that could be validated in the chimeric mouse model. Overall, this is a nice summary of mechanisms of HBV-related hepatocarcinogenesis with some discussions of the potential usefulness of the human hepatocyte chimeric mice model to confirm previous findings. I suggest the author include some discussions on the mice model with the humanized immune system, which might be helpful for the studies of interplays between the immune system and HBV-related HCC.

Author Response

I thank you for your constructive comment. I have discussed about the dual chimeric mouse model (Lines 110–122).

Reviewer 2 Report

The aim of this manuscript is to discuss the usefulness of a humanized mouse model, for the analysis of HBV-associated hepatocarcinogenesis.

Even if the manuscript provides an organic overview, with a densely organized structure and based on well-synthetized data, there are aspects to be mentioned, to make the article fully readable. For these reasons, the manuscript requires minor changes.

Please find below an enumerated list of comments on my review of the manuscript:

LINE 33: Hepatitis B Virus infection is a global health problem and a major cause of acute anche chronic liver disease. This issue is higlighted by different and recent studies (see, for reference: Long-term immune protection against HBV: associated factors and determinants – 2021), which also discuss the importance of a long-term immune protection and the immonogenicity of HBV vaccine.

LINE 289: As regards the epigenetic modifications associated to carcinogensis, recent findings on DNA methylation provide new insight into the immune dynamics, underlying HBV-infection (see, for reference: The signature of HBV-related liver disease in peripheral blood mononuclear cell DNA methylation), suggesting a linkage between HBV-related liver fibrosis and alterated CpG mathylation.

To sum up, the topic is timely and call for attention. Overall, the manuscript requires minor changes (as mentioned). I would accept the manuscript, if the comments are addressed properly.

Author Response

1. LINE 33: Hepatitis B Virus infection is a global health problem and a major cause of acute and chronic liver disease. This issue is highlighted by different and recent studies (see, for reference: Long-term immune protection against HBV: associated factors and determinants – 2021), which also discuss the importance of a long-term immune protection and the immonogenicity of HBV vaccine.

Response: I thank you for your constructive comment. I have mentioned this point in Lines 35–39.

2. LINE 289: As regards the epigenetic modifications associated to carcinogensis, recent findings on DNA methylation provide new insight into the immune dynamics, underlying HBV-infection (see, for reference: The signature of HBV-related liver disease in peripheral blood mononuclear cell DNA methylation), suggesting a linkage between HBV-related liver fibrosis and alterated CpG methylation.

Response: I thank you for your constructive comment. I checked the suggested and added the necessary information in Lines 302–306.
